



# Investigating the impact of exit effects on solute transport in macropored porous media

Jérôme Raimbault[1], Pierre-Emmanuel Peyneau[1], Denis Courtier-Murias[1], Thomas Bigot[1], Jaime Gil Roca[2], Béatrice Béchet[1], and Laurent Lassabatère[3]

[1]GERS-LEE, Univ Gustave Eiffel, IFSTTAR, F-44344 Bouguenais, France
[2]Laboratoire Navier, Ecole des Ponts ParisTech, CNRS, Univ Gustave Eiffel, 6-8 avenue Blaise Pascal, 77455 Marne-la-Vallée, France
[3]Univ Lyon, Université Claude Bernard Lyon 1, CNRS, ENTPE, UMR5023 LEHNA, F-69518, Vaulx-en-Velin, France

**Correspondence:** Pierre-Emmanuel Peyneau (pierre-emmanuel.peyneau@univ-eiffel.fr)

**Abstract.** The effect of macropore flow on solute transport has spurred much research over the last forty years. In this study, non-reactive solute transport in columns filled by macropored porous media was experimentally and numerically investigated, and the emphasis was put on the study of exit effects, whose very existence is inherent to the finite size of any experimental column. We specifically investigated the impact of the presence of a filter at the column outlet on water flow and solute

transport in macropored systems. Experiments involving breakthrough measurements and magnetic resonance imaging (MRI) showed that solute transport displayed some significant non-unidirectional features, with a strong mass exchange at the interface between the macropore and the matrix. Fluid dynamics and transport simulations indicated that this was due to the non-unidirectional nature of the flow field close to the outlet filter. The flow near the exit of the column was shown to be strongly impacted by the presence of the outlet filter, which acts as a barrier and redistributes water from the macropore to the matrix.

This impact was apparent on the breakthrough curves and the MRI images. It was also confirmed by computer simulations and could, if not properly taken into account, impede the accurate inference of the transport properties of macropored porous media from breakthrough experiments.

## 1 Introduction

Column experiments are frequently performed to study the transport of various contaminants in soils (De Matos et al., 2001;

Pang et al., 2002; Banzhaf and Hebig, 2016; Jin et al., 2000) or to fit experimental data with a transport model (Nielsen and Biggar, 1961; De Smedt and Wierenga, 1984; Cortis and Berkowitz, 2004). Broadly speaking, the general motivation shared by all these experiments is to study and to understand the transport processes occurring in the bulk of a porous medium in a simple and reproducible setting, by imposing a stationary flow along the axis of the column.

Experimentally, this task is more challenging than it might appear at first sight. The finite size of the column can impact

water flow and solute transport with, for instance, the existence of entrance/exit effects affecting the uniformity of the flow near the extremities of a column  (Starr and Parlange, 1977; Bromly et al., 2007). Similar issues have been underlined in chromatography (Guiochon, 2006; Farkas et al., 1994; Baur et al., 1988; Farkas et al., 1997; Shalliker et al., 2000; Broyles





et al., 1999; Gritti and Gilar, 2019) and in some fundamental studies of transport in porous media (Lehoux et al., 2016; Deurer et al., 2004; Greiner et al., 1997). The perturbations induced by the presence of the entrance and the exit ends of the column

can have a concrete incidence (e.g., flow disturbance and recirculation in the system reservoirs, additional solute dispersion). Consequently, the breakthrough curves (BTCs) may be affected by entrance/exit effects and may no longer reflect the intrinsic transport properties of the porous medium, but the transport properties of the whole experimental system (Lehoux et al., 2016; Schwartz et al., 1999; Starr and Parlange, 1977; James and Rubin, 1972). Several parts of the column device can impact the solute breakthrough (Giddings, 2002): upstream and downstream reservoirs, restrictions between the reservoirs and the tubes,

and frits or filters positioned at the inlet and outlet. These inert physical filtration devices are often employed to diffuse the incoming water flow evenly on the entrance face of the porous medium and to prevent porous medium particles from exiting and clogging the tubes downstream.

All these parts, located right before and/or right after the porous medium, may trigger disturbance of water flow and solute transport, especially when the porous medium under scrutiny is heterogeneous (Barry, 2009). Heterogeneous columns have in

particular been employed in transport studies motivated by questions raised by the complexity of water infiltration and mass transport in real soils. Real soils frequently contain macropores (Beven and Germann, 2013), which are large and continuous openings known to be involved in the rapid displacement of water and chemical substances, and various breakthrough experiments have been performed to study the role played by single macropores embedded in porous medium (Allaire et al., 2009). Unsaturated conditions being difficult to sustain in a well-controlled fashion, and the effect of macropores being expected to

culminate when they are activated (i.e., water-saturated), many results have been obtained from macropored columns operated in the saturated regime, with different artificial systems: packed soils containing constructed macropores, macropored sandy media, glass bead packings crossed by a macropore, etc. (Allaire et al., 2009; Li and Ghodrati, 1997; Ghodrati et al., 1999; Lamy et al., 2009; Batany et al., 2019). However, to our knowledge, the potential impact of entrance/exit effects on solute transfer through macropored porous media has never been investigated so far.

This paper aims to demonstrate the significant influence of the presence of an outlet filter on water flow and non-reactive solute transport within an artificial macropored system (the inlet filter was always set in place to prevent any clogging of the macropore). Using a combination of breakthrough experiments, MRI monitoring and computer simulations, we show that water flow and non-reactive solute transport in macropored porous media are strongly affected by the presence of a filter at the end of the column. This filter influences the velocity field in a sizable fraction of the macropored porous medium and strongly

impacts the transport of solute in the macropored porous medium and its elution at the outlet.

## 2 Materials and methods

### 2.1 Porous media and columns

We have constructed experimental columns filled with Hostun sand (HN 0.6/1.6, Sibelco, France). Before any experiment, the sand was sieved at $0.5\,\mathrm{mm}$ with a stainless mesh sieve. The sand was first washed with a $2\,\mathrm{mol\,L^{-1}}$ nitric acid solution, obtained

by diluting nitric acid $65\%$ (Emsure, Millipore) in ultrapure water (Milli-Q Integral 3 Water Purification System, Millipore).





The sand was then rinsed twice with ultrapure water and neutralized with a $0.1 \, \mathrm{mol \, L^{-1}}$ potassium hydroxide solution obtained by diluting $1 \, \mathrm{mol \, L^{-1}}$ potassium hydroxide (Titripur, Millipore) with ultrapure water. Afterwards, the sand was rinsed several times with ultrapure water until the pH of the solution reached the pH of the rinsing solution. Finally, the sand was dried at $105 \, ^\circ\mathrm{C}$ during 24 h and then stored in a plastic container. The particle size distribution of the sand was measured by laser

diffraction (Mastersizer 3000, Malvern). It ranged between 0.30 mm and 1.10 mm, with a median particle diameter equal to 1.0 mm. Pore size distribution was also characterized by X-ray tomography (SkyScan 1275 micro-CT, Bruker) and the median pore diameter was found to be equal to $d_{50} = 0.38$ mm.

Two kinds of hollow cylinders were used as macropores. They were 3D printed (Form 1+, Formlabs) using a photoreactive resin (Clear Resin, Formlabs). The hollow cylinders had an inner diameter $\mathrm{id}_m = 3.0$ mm, an outer diameter $\mathrm{od}_m = 5.0$ mm,

and a height of 15.0 cm. The first hollow cylinder was plain (no holes), whereas the second one was perforated with 0.5 mm diameter holes resulting in a 25% surface porosity. These two hollow cylinders were used to model impermeable and permeable macropores, respectively: water could flow and solute could cross the boundaries of the perforated hollow cylinder, whereas the plain one was impermeable to water flow and solute transfer.

We used XK 50/30 (Cytiva) columns to pack the porous media. The inner diameter of each column was equal to $d_{\mathrm{col}} = 5.0$

cm and their height was equal to $L = 15.0$ cm. The macropored columns were set up by inserting the hollow cylinders along the axis of the columns. Then, the Hostun sand was slowly poured around and dry packed thanks to gentle manual vibrations. Once filled with sand, each column was saturated during 2 h with carbon dioxide, a gas much more soluble in water than air. The column was then slowly water-saturated with a conditioning solution. Then, it was rinsed with 12 pore volume of the same conditioning solution at different flow rates (from 0.5 to $3.0 \, \mathrm{mL \, min^{-1}}$) to stabilize the pH and the electrical conductivity.

Mesh filters (Net Rings, Cytiva) adapted to the XK 50/30 columns with 10 $\mu$m pores were positioned just before and right after the porous medium. The exit effect, which is the focus of this study, was studied by removing the outlet filter for some of the experiments.

Three experimental columns (denoted A, B and C) were prepared according to the aforementioned methodology: column A is a homogenous control column, without any macropore, column B contains a perforated hollow cylinder along its axis acting

as a permeable macropore and column C contains a plain hollow cylinder along its axis acting as an impermeable macropore. The columns are depicted in Fig. 1. The pore volume of each column was estimated by weighting the column before and after saturation. The values were 119.5, 116.6 and 120.2 mL for columns A, B and C, respectively.

## 2.2 Aqueous solutions

We used two conditioning solutions and two tracer solutions. A $1.0 \times 10^{-4} \, \mathrm{mol \, L^{-1}}$ potassium nitrate ($KNO_3$) solution was

used as the first conditioning solution. The first tracer solution was a $1.0 \times 10^{-2} \, \mathrm{mol \, L^{-1}}$ potassium nitrate solution. Both solutions were prepared by dissolving solid potassium nitrate (Emsure, Millipore) in ultrapure water. The conditioning and tracer solutions had an electrical conductivity of $\sigma_0 = 0.01 \, \mathrm{mS \, cm^{-1}}$ and $\sigma_1 = 1.19 \, \mathrm{mS \, cm^{-1}}$, respectively. These solutions were used for the determination of the BTCs.





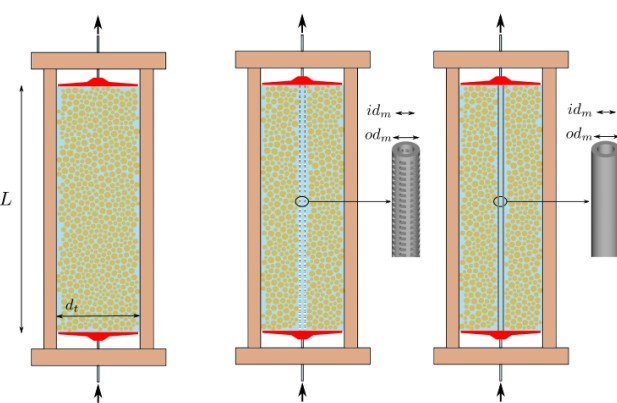

**Figure 1.** Columns used for the injection of solutes: homogenous control column A (left), macropored columns with the perforated hollow cylinder B (center) and the plain hollow cylinder C (right). The liquid distribution and collection systems are drawn in red.

The second conditioning and tracer solutions were prepared by dissolving gadolinium(III) chloride hexahydrate (Sigma-Aldrich) in ultrapure water. The second conditioning solution was a $1.0 \times 10^{-4}\,\mathrm{mol\,L^{-1}}$ $\mathrm{GdCl_3}$ solution (electrical conductivity $\sigma_0 = 0.03\,\mathrm{mS\,cm^{-1}}$) and the second tracer solution was a $1.0 \times 10^{-2}\,\mathrm{mol\,L^{-1}}$ $\mathrm{GdCl_3}$ solution ($\sigma_1 = 2.86\,\mathrm{mS\,cm^{-1}}$). We used these solutions for visualizing water flow and $\mathrm{Gd^{3+}}$ transport within the column by MRI (see Sec. 2.4). Complementary BTCs were also measured with this second set of solutions.

All the solutes were considered to behave as tracers, i.e. non-reactive chemical species following the water flow without any sorption, neither to the particles of the porous media nor to the walls of the hollow cylinder.

## 2.3 Breakthrough experiments

Columns were arranged vertically and solutions were injected from the bottom to the top, using a peristaltic pump (Ismatec ISM834A) connected to the injection system of a ÄKTAprime device (Cytiva) with polyether ether ketone (PEEK) tubings having a $0.75\,\mathrm{mm}$ inner diameter (Cytiva) and capillary tubing with an inner diameter of $1.55\,\mathrm{mm}$ included in the adapters of the XK 50/30 column. This low-pressure liquid chromatography system was used to continuously monitor electrical conductivity, pH, UV absorbance and temperature at the outlet of the column.

Each breakthrough experiment began with the injection of more than 2 pore volumes of conditioning solution to stabilize the pH and the electrical conductivity measured at the outlet of the column. Then, $5\,\mathrm{mL}$ of tracer solution were injected. The flow rate was set at $Q = 3\,\mathrm{mL\,min^{-1}}$, corresponding to a mean Darcy velocity $q = 4Q/(\pi d_{\mathrm{col}}^2)$ equal to $0.15\,\mathrm{cm\,min^{-1}}$. Each breakthrough experiment was triplicated. The relative concentration was determined from the measurement of the electrical conductivity $\sigma$ at the outlet as follows,

$$C = \frac{\sigma - \sigma_{\min}}{\sigma_1 - \sigma_{\min}}, \tag{1}$$





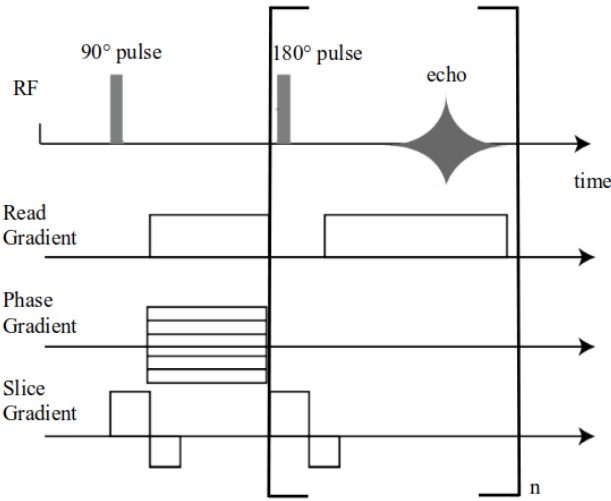

**Figure 2.** MRI sequence employed for image acquisition.

where $\sigma_{\min}$ stands for the minimum electrical conductivity obtained when the conditioning solution is injected and $\sigma_1$ is the electrical conductivity of the tracer solution.

## 2.4 Magnetic resonance imaging

The transport of $Gd^{3+}$ within a column, which is an opaque three-dimensional system, was monitored with a vertical nuclear magnetic resonance spectrometer (DBX 24/80 Bruker), operating with a $0.5$ T static magnetic field (20 MHz $^1$H frequency), and equipped with a birdcage radio-frequency coil delimiting a measurement zone of 20 cm in diameter and 20 cm in height. Due to its paramagnetic properties, $Gd^{3+}$ is known to be an excellent MRI contrasting agent (Pyykkö, 2015), and has already been used to study solute transport in soils (Haber-Pohlmeier et al., 2017).

As for the breakthrough experiments, the $Gd^{3+}$ solution was injected at $3.0\,\mathrm{mL\,min^{-1}}$ in the column from the bottom with a peristaltic pump connected to the ÄKTAprime device. The sole difference between the classical and the MRI monitored breakthrough experiments was that in the latter case, the connecting tubes were longer (10 m before the entrance and 10 m after the exit of the column), so that the injection system was outside the MRI setup.

Two-dimensional MRI vertical slices of 6 mm thickness, encompassing the axis of the column, were taken at different times during the injection of the solute. Each image had $128 \times 64$ pixels and was acquired in 3 min 55 s. The field of view was $19\,\mathrm{cm} \times 5.5\,\mathrm{cm}$, providing a spatial resolution of $1.48$ mm/pixel $\times 0.85$ mm/pixel. A multi-spin multi-echo (MSME) sequence based on a succession of 16 echoes was used, with an echo time $T_E = 7.4$ ms, and a recycle delay $T_R = 1.2$ s. This multi-echo sequence, schematized in Fig. 2, has been employed to improve the signal-to-noise ratio, without increasing the measurement time, as the measured 16 echoes were added to produce a single two-dimensional image (Zhou et al., 2019), thus preventing direct concentration quantification. Moreover, due to the short recycle delay used to keep measurement time below 4 min,





the resulting signal depends simultaneously on the spin-lattice relaxation time $T_1$ and on the spin-spin relaxation time $T_2$, thus complicating quantification through a simple comparison with a reference. The MRI images can nevertheless be used to evaluate where $Gd^{3+}$ is present within the column.

## 2.5 Computer simulations

Numerical simulations were performed with COMSOL Multiphysics (version 5.4), a commercial finite element software. COMSOL Multiphysics was used to define the geometry of the problem, to generate the computational mesh and to solve the partial differential equations governing the fluid flow and the non-reactive transport of the solute, with the specified initial and boundary conditions.

To simulate column B, we developed a 2D axisymmetric geometric model (15.0 cm length and 2.5 cm radius) with two regions: one for the sandy matrix and another one for the macropore. The filters were represented as $10\,\mu\text{m}$ thick porous media. The geometry of the inlet and outlet reservoirs was also taken into account in the numerical model.

The mesh was automatically built by COMSOL Multiphysics. It was adapted to the geometry previously defined with an increase in node density at the interfaces between subdomains and in small subdomains (like the filters or the reservoirs). We checked that the numerical results remained unaffected when the mesh was refined.

The stationary flow of the carrier liquid within the column was described by the Stokes equations in the macropore (free region) and by the Brinkman equations in the surrounding porous media (Guyon et al., 2015). The Stokes equations read:

$$\begin{cases} -\nabla p + \mu \triangle \mathbf{u} = 0 \\ \nabla \cdot \mathbf{u} = 0 \end{cases}. \tag{2}$$

$p$ stands for the liquid pressure, $\mu$ for the dynamic viscosity of the liquid and $\mathbf{u}$ for the velocity field of the liquid.

In the surrounding porous medium and in the filters, the Brinkman equations were used to model the liquid flow since momentum transport induced by shear stresses is of importance at the interface between the macropore and the porous matrix (Ochoa-Tapia and Whitaker, 1995). These equations extend Darcy's law to describe the dissipation of the kinetic energy by viscous shear and read:

$$\begin{cases} -\nabla p + \mu \phi^{-1} \triangle \mathbf{u} - \mu \kappa^{-1} \mathbf{u} = 0 \\ \nabla \cdot \mathbf{u} = 0 \end{cases}. \tag{3}$$

$\phi$ is the porosity and $\kappa$ the permeability of the porous matrix.

The transport of the non-reactive solute was modeled with the advection-diffusion equation in the macropore and the advection-dispersion equation in the porous medium. Both equations can be written as:

$$\frac{\partial c}{\partial t} = \underline{\underline{\mathbf{D}}} \cdot \nabla^2 c - \mathbf{u} \cdot \nabla c. \tag{4}$$

In the macropore, $\underline{\underline{\mathbf{D}}}$ denotes the isotropic tensor $D_0 \underline{\underline{\mathbf{I}}}$, $D_0$ being the molecular diffusion coefficient of the solute and $\underline{\underline{\mathbf{I}}}$ the second-order identity tensor. In the porous matrix and in the filters, $\underline{\underline{\mathbf{D}}}$ denotes the transversely isotropic tensor $D_L \hat{\mathbf{u}} \otimes \hat{\mathbf{u}} +$





$D_T(\underline{\underline{\mathbf{I}}} - \hat{\mathbf{u}} \otimes \hat{\mathbf{u}})$, where $D_L$ is the longitudinal coefficient of dispersion, $D_T$ the transversal coefficient of dispersion, $\hat{\mathbf{u}}$ the normalized vector $\mathbf{u}/|\mathbf{u}|$ and the symbol $\otimes$ denotes the tensor product. $D_L$ and $D_T$ combine the effects of both molecular diffusion and mechanical dispersion and can be written as follows (Bear, 1988):

$$\begin{cases} D_L = \lambda_L |\mathbf{u}| + \tau\, D_0 \\ D_T = \lambda_T |\mathbf{u}| + \tau\, D_0 \end{cases}. \tag{5}$$

$\tau$ the tortuosity of the porous medium, $\lambda_L$ the longitudinal dispersivity and $\lambda_T$ the transverse dispersivity. Moreover, we assumed for the sake of simplicity that the transverse dispersivity was equal to one-tenth of the longitudinal dispersivity, $\lambda_T = \frac{1}{10}\lambda_L$ (Zech et al., 2018).

Solving Eqs. 2, 3 and 4 requires the knowledge of the hydraulic and transport properties of the porous medium and the filters. Regarding hydraulic properties, the permeability of the sand was evaluated with the Kozeny-Carman equation (Guyon

et al., 2015),

$$\kappa = \frac{\phi^3 d_g^2}{180\,(1-\phi)^2}, \tag{6}$$

$d_g$ being the mean diameter of the grains. In order to fit the experimental BTCs, $d_g$ was taken equal to $0.57\,\mathrm{mm}$ (a value rather close to $d_{50} = 0.38\,\mathrm{mm}$ determined by X-ray tomography), yielding for the sand a permeability of $2.6 \times 10^{-10}\,\mathrm{m}^2$. The filter was modeled as thin porous slab periodically perforated by square holes of length $a = 10\,\mu\mathrm{m}$. The surface porosity of the slab

has been taken equal to $\phi_{\mathrm{filter}} = 25\%$. According to Bruus (2007), the permeability of a channel with a square cross-section of side length $a$ is equal to

$$\kappa_{\mathrm{sq}} = \frac{a^2}{12}\left[1 - \frac{192}{\pi^5}\sum_{n=0}^{+\infty}\frac{1}{(2n+1)^5}\tanh\left(\left(n+\frac{1}{2}\right)\pi\right)\right]. \tag{7}$$

Numerically, this yields $\kappa_{\mathrm{sq}} \simeq 3.5 \times 10^{-2}\,a^2$. The permeability of the filter was taken equal to $\kappa_{\mathrm{filter}} = 3.5 \times 10^{-2}\,a^2\,\phi_{\mathrm{filter}} = 8.7 \times 10^{-13}\,\mathrm{m}^2$. The longitunal dispersivity of the porous matrix was taken equal to the mean pore size, i.e. $0.40\,\mathrm{mm}$ for the

sand and $10\,\mu\mathrm{m}$ for the filters, and the tortuosity was set equal to $1$.

As for the boundary conditions, a given flow rate was imposed at the inlet of the column and a uniform pressure was imposed at the outlet. For the solute, we considered a concentration flux condition at the entry of the system. To model the injection of a 5 mL volume of tracer solution, we set the concentration flux to 1 during the first 5 mL of injected solution and to 0 afterwards. Since Eq. 4 is linear in $c$, the concentration calculated this way is equal to the normalized concentration, whatever the genuine

value of the physical concentration of the tracer solution at the inlet of the column.

The flow field within the columns, the temporal evolution of solute concentration maps and numerical BTCs were then computed by solving numerically Eqs. 2, 3 and 4.



## 3 Results and discussion

### 3.1 Breakthrough curves

The normalized concentrations measured at the outlet of the columns A, B and C, plotted as a function of the number of pore volumes (PV), are depicted in Fig. 3. For the experiments reported in the present section, $KNO_3$ solutions have been used as conditioning and tracer solutions. We remind that PV is equal to $Qt/V_0$, where $V_0$ denotes the pore volume of the column and $t$ the elapsed time since the beginning of the injection of the tracer solution. We conducted three breakthrough experiments for each column, and the corresponding error bars are shown in Fig. 3.

The BTCs of the homogeneous column (column A) are displayed in Fig. 3a. They have been measured in the presence and in the absence of the outlet filter and are both slightly asymmetric bell-shaped curves, a standard shape for columns filled with homogeneous porous media. The two BTCs are nearly indiscernable, which implies that the outlet filter has no impact on solute transport in the homogeneous case.

The results are very different for the two macropored columns, B and C. The BTCs measured at the outlet of column B
(perforated macropore) are depicted in Fig. 3b, in the presence (blue curve) and in the absence (red curve) of an outlet filter. The two BTCs share some features, such as the existence of two distinct peaks. The first peak is very asymmetric, with a short ascent followed by a long tail. Breakthrough starts for small values of the number of pore volumes ($PV \leq 0.02$) and the maximum of the first peak is reached for $PV \simeq 0.05$. The second peak is much more symmetric and reaches its maximum after more than 3 pore volumes. However, the two BTCs differ with respect to the position of the maximum of the second peak:
it is located at $PV \simeq 3.2$ when the outlet filter is present and at $PV \simeq 3.6$ without any outlet filter. This discrepancy entails that the mean residence time associated with the second peak is affected by the presence of the outlet filter. Besides, this mean residence time is directly related to the mean longitudinal pore velocity of the solute giving rise to the second peak of the BTCs. Consequently, the difference in PVs means that the flow within the column is affected by the presence of the outlet filter.

The analysis of the BTCs of column B gives further insight into the characteristics of the flow field within this column.
The decrease of the first peak is surprisingly slow. The normalized concentration remains above zero at least up to $PV = 1.5$, whereas the volume of the macropore is only $1\%$ of the total pore volume of the system. The slow decrease of the normalized concentration measured at the outlet of column B thus hints at the existence of a substantial solute transfer between the porous matrix and the macropore.

We performed the same kind of experiments by replacing the perforated hollow cylinder used in column B by a plain one to
investigate the possible occurrence of such a transfer. The BTCs measured at the outlet of column C are depicted in Fig. 3c, in the presence (blue curve) and in the absence (red curve) of the outlet filter. The shape of the BTCs is less affected by the presence of the outlet filter than for column B. The comparison of Fig. 3b and Fig. 3c shows that the decrease of the first peak is much more pronounced for column C. In this column, by construction, water and solute exchange is prohibited between the macropore and the porous matrix. Thus, the solute having entered the column through the macropore (respectively, through
the porous matrix) remains in the macropore (respectively, in the porous matrix) throughout its transport within the column. Accordingly, the mass balances corresponding to both peaks are directly related to the fractions of water flowing through

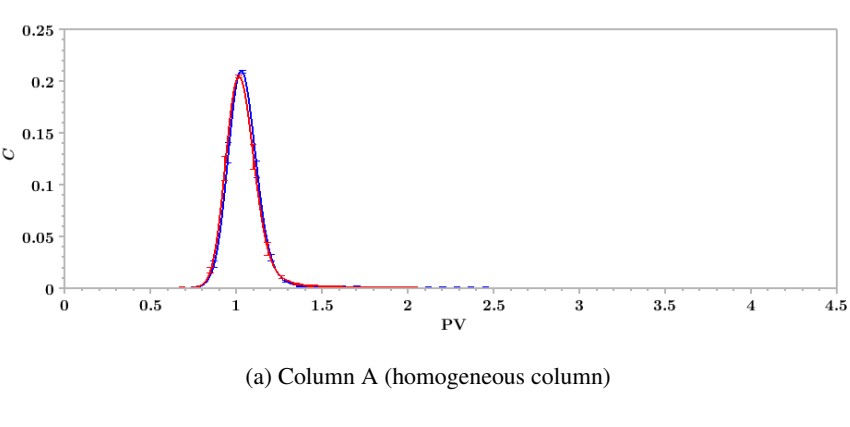

(a) Column A (homogeneous column)

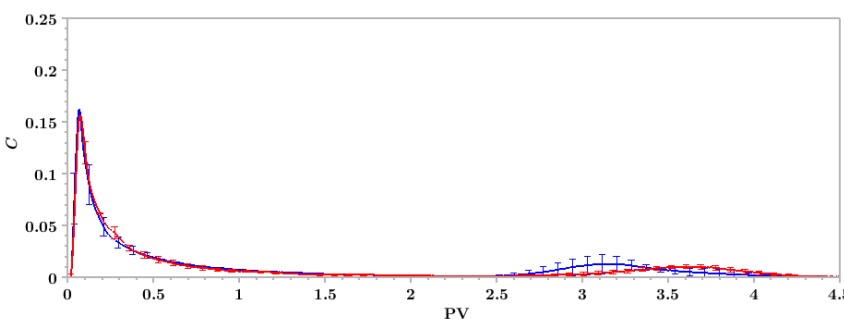

(b) Column B (heterogeneous column with a perforated macropore)

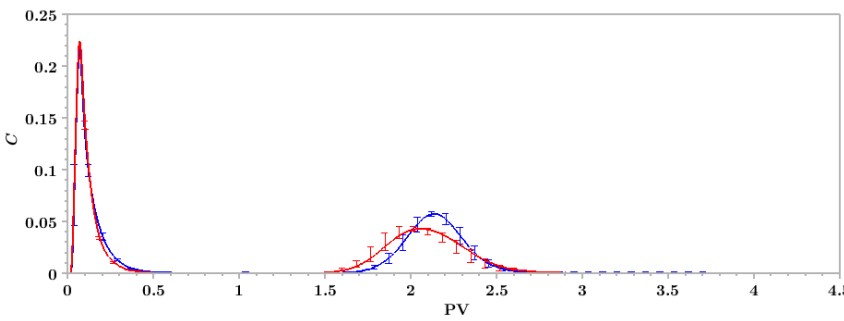

(c) Column C (heterogeneous column with a plain macropore)

**Figure 3.** Breakthrough curves, showing the normalized concentration $C$ as a function of the number of pore volumes PV, measured at the outlet of columns A, B and C, in the presence (blue curve) and in the absence (red curve) of an outlet filter. Each breakthrough experiment has been conducted three times.

the macropore and through the matrix. This is not the case for column B, whose macropore is perforated, and that can thus experience some solute transfer between the macropore and the surrounding matrix. The difference between the BTCs of columns B and C shows that such solute transfer does occur and explains the long tail associated with the first peak measured

at the outlet of column B. Moreover, in contrast with column B, the area below the first peak of column C is less than half of





the total area below the BTC, providing further information on the extent of solute transfer between the macropore and the porous matrix in column B.

Finally, for column C, the second peaks reach their maxima for rather similar values of PV (PV $\simeq 2.1$ in the presence of the outlet filter and PV $\simeq 2.0$ in its absence). These values are smaller than those observed for column B, which means that the mean longitudinal pore velocity of the solute associated with the second peak is significantly greater in column C than in column B. It shows that, besides the alteration of water and solute exchange at the interface between the macropore and the porous matrix, the presence of a plain macropore also modifies the flow field within the column.

## 3.2 MRI monitoring of $Gd^{3+}$ transport

A set of successive two-dimensional MRI images illustrating the time evolution of $Gd^{3+}$ presence within column B is shown in Fig. 4b and 4c. These images were taken over the course of the injection of the $GdCl_3$ tracer solution. Due to magnetic field heterogeneity, the images are deformed near the entrance and the exit of the column, leading to the distortion of the column lateral boundary on the images. However, this imperfection does not hinder their qualitative exploitation. Moreover, we also carried out additional breakthrough experiments, this time with $GdCl_3$ conditioning and tracer solutions (see Fig. 4a). The general features of the $GdCl_3$ BTCs are similar to those of the $KNO_3$ BTCs (displayed in Fig. 3b), discussed in the previous section.

In the successive two-dimensional images displayed in Fig. 4b and 4c, the grey level of each pixel is sensitive to two parameters, the local porosity (since the column is saturated, the higher the local porosity, the higher the quantity of water in a given small volume and the higher the MRI signal) and the $Gd^{3+}$ local concentration (the higher the concentration of $Gd^{3+}$, the lower the MRI signal) (Haber-Pohlmeier et al., 2017). In the first image of Fig. 4b, before the beginning of the injection, the macropore, where the local porosity is equal to 1, appears in light grey, whereas the surrounding porous matrix, which local porosity $\simeq 0.4$, appears in medium grey. In the following images ($PV_1 - PV_8$), some solute is present in the column and its local concentration is positively correlated with the pixel level of darkness.

The transport of $Gd^{3+}$ within the porous matrix can easily be observed, both in the presence (Fig. 4b) and in the absence of the outlet filter (Fig. 4c). We start the discussion by the case where the outlet filter is present (Fig. 4b). At $PV_1$, a dark cone appears just before the outlet filter. Meanwhile, a $Gd^{3+}$ front appears in the porous matrix surrounding the macropore at the bottom of the column. Then, as can be seen at $PV_2$, the cone extends downwards and laterally towards the lateral boundary of the column. Moreover, the front visible in the porous matrix moves upwards. Subsequently, a brighter zone appears at the center of the cone (at $PV_3$), then invading the whole conical region, except along the boundaries of the cone which remain slightly dark (see $PV_4$ image). Meanwhile, the $Gd^{3+}$ front continues its ascent into the porous matrix. When the $Gd^{3+}$ front approaches the column outlet and reaches the tip of the cone, it starts to distort (at $PV_5$ and $PV_6$) before disappearing (at $PV_7$ and $PV_8$). During this last stage, the front is made of two very distorted parts that move away from the macropore and the central part of the column.

In the absence of the outlet filter (Fig. 4c), the situation differs. The MRI images displayed in Fig. 4c show that, in the matrix, the elution front moves upwards with a nearly horizontal shape, except for a small distortion close to the lateral boundary of





the columns. This small deformation of the front is probably due to the existence of a slight preferential flow along the lateral boundary of the column (also visible when a filter is present, cf. Fig. 4b). Moreover, the horizontal shape of the front is altered when it approaches the exit of the column (see $PV_5 - PV_8$ images), but to a lesser extent than when the outlet filter is present. We can also notice that the macropore region appears slighly darker at the beginning of the injection ($PV_1$ and $PV_2$ images): this change of color is likely related to the transfer of $Gd^{3+}$ in the macropore. Furthermore, in the absence of the outlet filter,

no conical shape appears close to the outlet of the column.

Thus, the strong impact related to the presence of a filter at the outlet of column B, visible on the BTCs and already discussed in the previous section, is also clearly visible in the MRI experiments. When such a filter is present, the time evolution of the solute concentration map within the column is rather complex and displays some marked two-dimensional features. The computer simulations presented in the following section will be helpful to gain a better understanding of the flow processes

and their impact on non-reactive solute transport occurring in column B, both in the presence and in the absence of the outlet filter.

### 3.3 Finite element computations

We solved numerically Eqs. 2, 3 and 4 in a two-dimensional axisymmetric domain representing column B, with and without an outlet filter. The modeled BTCs (Fig. 5a) and resident normalized concentration (Figs. 5b and 5c) are in good agreement with

the experimental data presented in Fig. 4. Moreover, the most eye-catching feature related to the influence of the outlet filter on the experimental BTCs, which is its influence on the position of the second peak, is well reproduced by the numerical BTCs displayed in Fig. 5a. As in the breakthrough experiments (see Fig. 3b), the second peak shifts leftwards, its maximal value increases and its width decreases when the outlet filter is added. On the downside, the modeling of the first peak appears to be challenging and we did not succeed in reproducing quantitatively this portion of the experimental BTCs. Small geometrical

details close to the entrance of the column have a sizable effect on the first peak of the numerical BTCs and make it difficult to go beyond the qualitative agreement that we nevertheless highlighted.

The numerical concentration maps are in good qualitative agreement with the MRI images. In the presence of the outlet filter (see Fig. 5b), a conical shape rich in solute appears right after the beginning of the injection close to the exit of the column, as observed in the MRI experiments (cf. Fig. 4b). In addition, the model predicts the progressive fading of this conical shape

with the temporary persistence of a solute-rich region along its boundaries (see $PV_3$ image of Fig. 5b). It also reproduces the upwards transport of the solute front within the porous matrix: the front in the porous matrix is nearly horizontal during the initial stage of the transport (images $PV_1 - PV_4$ of Fig. 5b), before being strongly distorted while approaching the exit of the column (images $PV_5 - PV_8$ of Fig. 5b), in good qualitative agreement with the images acquired by MRI (cf. Fig. 4b).

The numerically computed concentration maps change significantly when the outlet filter is removed, but the agreement

between the calculated maps and the MRI images is still good. In the absence of the outlet filter, the solute remains much more located into the macropore at the beginning of the injection, with only a slight diffusion in its surroundings (see images $PV_1 - PV_3$ of Fig. 5c). Moreover, no conical region appears in the vicinity of the exit of the column. Afterwards, the model predicts the upwards movement of the solute front within the porous matrix, without any distortion and with a progressive exit through



the column outlet (images $PV_4 - PV_8$ of Fig. 5c). This pattern is similar to that observed by MRI (Fig. 4c). The sole perceptible
difference is that after $PV_4$, the front is curved upward in Fig. 4c, whereas it remains almost flat in the computer simulations.
We believe that this may be due to the existence of a small preferential flow along the lateral boundary of the experimental
column.

The good overall agreement between the numerical results and the observed data allows us to conclude on the way the solute
is transfered through column B and to explain the effect of the outlet filter. Without the outlet filter, the solute enters into the
macropore and the matrix and is then transported through these subdomains with a very moderate solute exchange between
these subdomains. Only a slight solute spread is visible in the upper part of the macropore (images $PV_1 - PV_3$ of Fig. 5c).
Solute concentration maps are similar in the lower half of the column, whatever the presence of an outlet filter, but drastic
changes occur in the upper half of the column depending on the presence of such a filter. The outlet filter triggers a significant
solute exchange between the macropore and the matrix, resulting in the appearance of a conical region rich in solute (images
$PV_1 - PV_3$ of Fig. 5b). Meanwhile, a fraction of the solute is transported through the matrix, and the corresponding front
remains nearly horizontal until its gets sufficiently close to the column outlet. Then, this front experiences a distortion and
moves towards the column lateral boundary.

To summarize, the outlet filter routes a fraction of the solute transiting through the macropore to the matrix before the exit
of the column. The presence of the filter also implies that the solute transported through the matrix avoid the macropore and
the central part of the column when approaching the column outlet.

The effect of the outlet filter on solute transport results from its effect on the flow. We analyzed the water flow field to better
understand how the presence of the outlet filter modifies the flow and thus impacts solute transport. Various features of the
flow field are depicted in Fig. 6. From the analysis of the streamlines (Figs. 6a and 6d), the velocity magnitude maps (Figs. 6b
and 6e), and the radial component of the velocity field at the interface between the macropore and the matrix (Figs. 6c and
6f), it is clear that the flow fields, with and without the outlet filter, are similar in the lower half of the column and strongly
differ in the upper half. The outlet filter triggers a divergence of streamlines from the macropore to the matrix close to the
column outlet (Fig. 6a) and thus a water flux along this direction, as revealed by the positive radial component of the velocity
vector at the macropore/matrix interface in the upper half of the column (Fig. 6c). This divergence and the related water flux
across the macropore/matrix interface are responsible for the main features visible both in the MRI images (Fig. 4b) and the
numerical solute concentration maps (Fig. 5b). Indeed, water routes the solute from the macropore to the surrounding matrix
by advection. This flow pattern explains the conical shape associated to solute transport through the macropore (images $PV_1$
$- PV_3$ of Fig. 5b). The same divergence routes the solute transported through the matrix far away from the center of the
column and thus closer to the column lateral boundary. It explains the strong distortion experienced by the matrix front when
it approaches the exit of the column (images $PV_4 - PV_8$ of Fig. 5b). In the absence of the outlet filter, there is no longer any
streamline divergence near the exit of the column exit (Fig. 6d), as the water flux at the macropore/matrix interface vanishes in
the upper half of the column (Fig. 6f), yielding the solute to remain in either the matrix or the macropore (images $PV_1 - PV_8$
of Fig. 5c), except for the possible occurrence of transport by molecular diffusion (Batany et al., 2019).





With its very low permeability, the outlet filter acts as a thin layer impeding flow. The effects of embedded layers in macro-pored porous media were already discussed by many authors. For instance, Lassabatere et al. (2004) and Lamy et al. (2013)
showed that the amendment of geotextiles in soil columns is an efficient way to homogenize flow and then foster pollutant removal by the matrix. These authors hypothesized that the geotextiles acted as impeding layers and redistributed flow from high permeability conducting zones to lower permeability matrix zones. Even if, in our study, the filter was positioned at the end of the column, the same kind of behavior seems to occur. The low permeability of the filter act as a barrier to the preferential flow in the macropore and routes parts of the water and the solutes to the matrix.

The analysis of the flow field shows that the same homogenizing effect also occurs at the inlet. Indeed, in the presence of the outlet filter, symmetrical streamline distortion and flow field were obtained at the inlet and at the outlet of the column. Figure 6a shows the convergence to the macropore of some streamlines having entered the column through the porous matrix after the inlet filter, and the divergence to the porous matrix of some streamlines coming from the macropore before the outlet filter, an effect already observed in other MRI studies (Deurer et al., 2004; Greiner et al., 1997). The presence of the inlet filter
tends to homogenize the magnitude of the fluid velocity right after the filter. Farther from the inlet filter, when its influence on the flow is no longer felt, the streamlines become almost parallel to the axis of the column: because of the symmetry of the streamlines in the presence of the outlet filter, this is only visible in the middle of the column in this case (see Fig. 6a), whereas it is apparent in the upper half of the column in the absence of any outlet filter (see Fig. 6d). In the region where streamlines are straight and parallel to each other, the flow is fully developed and the velocity field is similar to the one one
would get in the entire column if the water flow was unidirectional and pressure-driven. In the vicinity of the inlet filter, where the flow is not yet fully developed, the carrier liquid velocity increases within the macropore as one goes from the bottom to the middle of the column and simultaneously decreases in the porous matrix because of the incompressibility of the liquid flow (see for instance Fig. 6e). However, despite the symmetry of the streamlines between the inlet and the outlet of the column when filters are present at both extremities, qualitatively, solute transport seems to be rather unaffected by the presence of an
inlet filter, whereas it is strongly impacted by the presence of the outlet filter: images acquired by MRI (Figs. 4b and 4c) and computer simulations (Figs. 5b and 5c) show that the solute front is nearly horizontal in the lower half of a macropored column containing a perforated macropore. The presence of the inlet filter probably alter the mass repartition between the macropore and the surrounding matrix, but it does not give rise to a distortion of the solute front like the one generated by the diverging flow pattern induced by the presence of the outlet filter.

## 4    Conclusions

In this study, we investigated the effect of an outlet filter on solute elution and on the time evolution of concentration maps in homogeneous and macropored columns, considering both perforated (i.e. permeable) and plain (i.e. impermeable) macropores. For this purpose, we combined i) column breakthrough experiments with tracer solutions ($KNO_3$ and $GdCl_3$); ii) MRI experiments to monitor $Gd^{3+}$ transport within columns and iii) computer simulations of water flow and non-reactive solute
transport.



While the breakthrough curve is unaffected by the presence of an outlet filter when the column is homogeneous, this is no longer true for macropored columns, especially when the macropore is permeable to water and solute fluxes. Computer simulations show that this effect on flow and solute transport results from the barrier effect played by the outlet filter: the closer the macropore to the outlet filter, the smaller the velocity of the carrier liquid in the macropore, which leads to a

strong divergence of the streamlines of the carrier liquid near the outlet filter. This entails a substantial transfer of water and a joint transport of solute of advective origin from the macropore to the surrounding matrix. Meanwhile, the solute conveyed through the porous matrix is transported away from the central part of the column when approaching the outlet filter. This very significant alteration of the flow before the outlet filter, which is responsible for some distinctive features of non-reactive solute transport within such columns, is obvious on the MRI images and the simulated concentration maps (occurrence of conical

shapes) and explains the influence exerted by the outlet filter on the breakthrough curves.

The numerical results show that the presence of filters (at the outlet, but also at the inlet) can impact the flow of the carrier liquid over a significant part of the column. The flow can display some substantial non-unidirectional features associated with entrance/exit effects. Such finite length effects are expected to be less pronounced as the ratio between the length of the column and its diameter increases, but increasing this ratio is not always an option (e.g. because it entails the use of a greater amount

of material and of stock solutions, or because the columns may have to be small due to experimental constraints).

This study shows that a simple one-dimensional transport model will not necessarily be appropriate, even when $L/d_{\mathrm{col}} = 3$ (ratio we have worked with in this study). Indeed, when it comes to fitting the experimental data, a good knowledge is required regarding i) the stationary flow within the system; ii) the effect of the various elements of the experimental apparatus on solute transfer between domains differing in their hydraulic properties. Our results shows that this knowledge is crucial for the

understanding of the outcome of transport experiments in heterogeneous columns and for the accurate inference of transport properties from breakthrough studies.

As emphasized in this work, for different experiments to be reliably exploited and compared, there is a need to report accurately the geometric features of the column and the boundary devices employed when performing transport experiments with heterogeneous media (frits or filters, reservoirs, incoming tubes, etc.). It may only be possible to relate transport parameters

to porous medium properties by taking into account the whole experimental apparatus employed. This issue requires a careful consideration of the potential impact of the geometry of the column and the additional boundary devices to draw some quantitative estimates from experimental data obtained with macropored columns. In any case, more in-depth studies devoted to this subject are certainly called for.

*Author contributions.*  JR: conducted the breakthrough experiments, contributed to the acquisition of MRI data and to the numerical mod-

eling, and wrote the manuscript; PEP: contributed to design the research and wrote the manuscript; DCM: contributed to the acquisition of MRI data and to design the research, and edited the manuscript; TB: carried out the preliminary computer simulations; JGR: performed the MRI experiments; BB: designed the research and edited the manuscript; LL: designed the research and wrote the manuscript.



*Competing interests.* No competing interest to declare.

*Acknowledgements.* This work was performed within the INFILTRON project supported by the French National Research Agency (ANR-17-CE04-010). We thank Pascal Moucheront and Benjamin Maillet for MRI assistance, David Hautemayou for 3D printing, and Martin Guillon and Nadège Caubrière for technical assistance with the experimental columns.





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




(a)

(b)

(c)

**Figure 4.** (a) $GdCl_3$ experimental BTCs measured at the outlet of column B with (blue curve) and without (red curve) an outlet filter. MRI images taken during the course of the elution as a function of the number of pore volumes PV for column B, with the outlet filter (b) and without the outlet filter (c). Pores volumes corresponding to the time average at which each image has been acquired are reported in (a). The $GdCl_3$ tracer solution was injected at the bottom of the column.





(a)

(b)

(c)

**Figure 5.** (a) Numerical BTCs calculated for a column with a permeable macropore in the presence (blue curve) and in the absence (red curve) of an outlet filter. Solute normalized concentration maps calculated over the course of the elution of the solute, in the presence of the outlet filter (b) and without the outlet filter (c). The scale of the colorbar is logarithmic. The number of pore volumes PV at which the maps have been computed are reported in (a). The solute is transported from the bottom to the top of the domain.



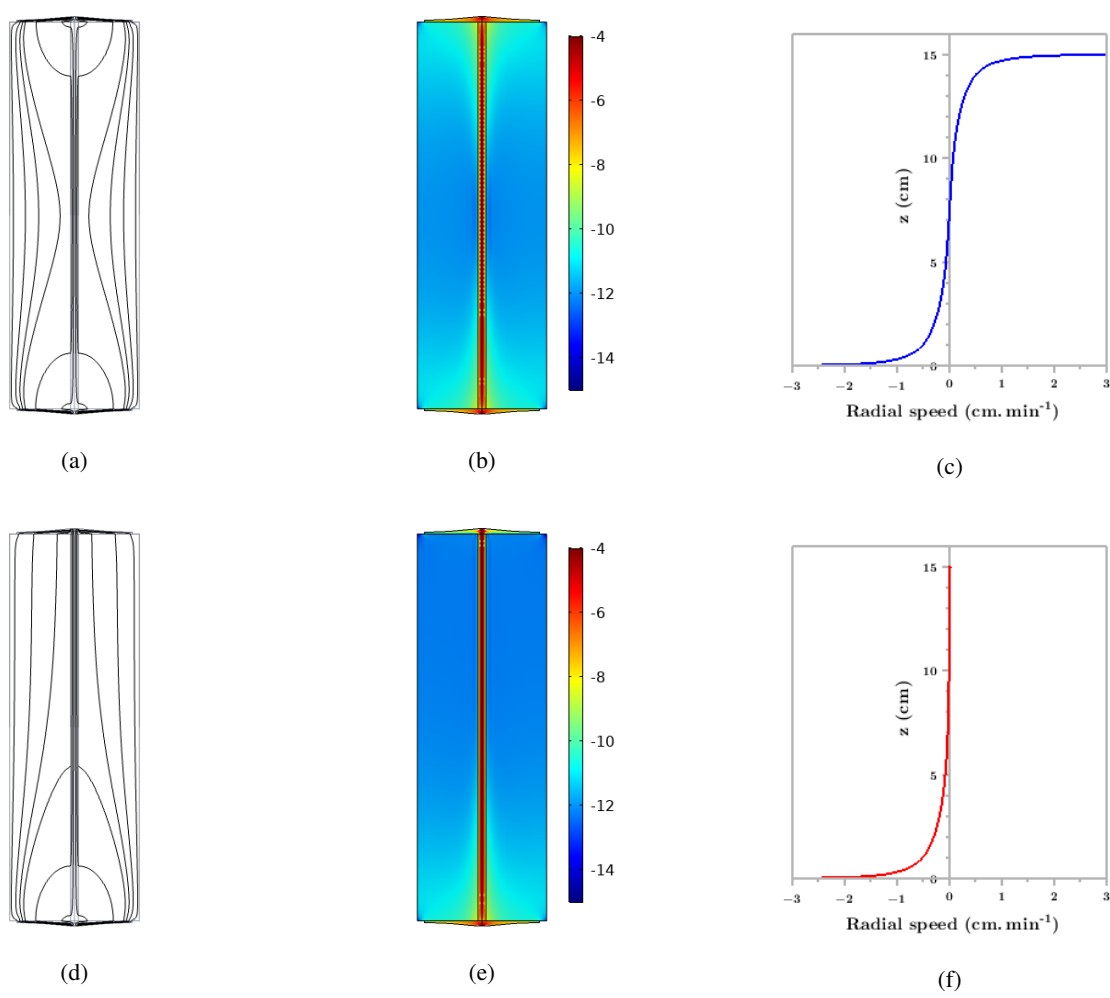

**Figure 6.** Various features of the flow field within a macropored column with a permeable macropore in the presence (first row) and in the absence (second row) of an outlet filter. (a) and (d): velocity field streamlines. (b) and (e): logarithmic map of the velocity magnitude of the carrier liquid expressed in $\mathrm{m\,s^{-1}}$. (c) and (f): radial component of the velocity vector on the matrix/macropore interface ($> 0$ when water flows from the matrix to the macropore and $< 0$ otherwise). The carrier liquid flows from the bottom to the top of the domain.