# Peer review of "Investigating the impact of exit effects on solute transport in macroporous media"

_Hydrology and Earth System Sciences, 2020_

## Referee Comment (RC1) · Anonymous Referee #1 · 5 Nov 2020

General Comments

The study aims to demonstrate an exit effect (outer filter) in solute transport in a soil column with artificial macropores. They utilize advanced methodologies such as breakthrough experiments, MRI, and Comsol Multiphysics software. The results showed that an exit effect occurs with a good agreement between the three independent methodologies. I agree with your findings and congratulate the researchers for the hard work. Indeed, the study shows an exit effect on solute transport. Therefore, the overall research question is answered.

Before publication, I suggest an in-depth English edition. I found some technical corrections (see below); however, I am sure that I drop some more. Finally, I would like the authors to answer the next general and specific comments.

Q1. You did a nice introduction to the problem. You highlight that column laboratory experiments are commonly used to study the transport of various contaminants in soils or fit experimental data with a transport model. Considering the second highlight (fit experimental data), we commonly fit the dispersion length for a non-reactive solute with BTCs without considering exit effects and then use it for model simulations. Given your results for columns B and C. Do you think that the magnitude of the fitted dispersion length using the blue or red curves in Figure 3, might be strongly different?. I mean, we should be worried about the current estimation of dispersion length without considering exit effect under laboratory conditions?. I think the discussion about this is important in your manuscript because you also mentioned this in the conclusions. You indeed found an exit effect with your study, but it can be irrelevant for the fitted parameter (e.g., dispersion length). Finally, it could be interesting to see the exit effect for an adsorbing solute with COMSOL to expand this study (optional).

Q2. You demonstrate an exit effect using a soil system with a high macropore diameter (3 mm) in a sandy matrix with a mean pore diameter of 0.58 mm (for some researches, that diameter is indeed a macropore itself). The 3 mm macropore connects perfectly to the top and bottom boundary of the column with the filter. Therefore, your system allows fast water flow and solute transport in both domains, which can be different from the standard description of macropore flow where the soil matrix has very low permeability, and macropores can terminate at different depths. Thus, I think your system is convenient for observing the exit effect because of the high differences in permeability between the filter and the bulk soil system. What happens with the exit effect if the macropore terminates in the soil matrix before reach the bottom filter? What happens with smaller macropore diameters where the flow velocity is lower than for 3 mm macropores?. These questions are important for analyzing your conclusions. Please check your conclusion from line 371 onwards; they cannot be general because your study is particular, perhaps, unrealistic.

Q3 Finally, in your conclusion (Ln 371), you mentioned that "Our results shows that this knowledge is crucial for the understanding of the outcome of transport experiments in heterogeneous columns and for the accurate inference of transport properties from breakthrough studies." As I mentioned in Q1, I think you did not demonstrate the last point. We do not know if the exit effect is critical for the accurate inference of transport properties from breakthrough studies with your study. From your study, we can know that the exit effect

happens in your particular setup (which is probably deviated for common heterogeneous soils with macropores).

Specific comments

Abstract: Please mention that your study is only under saturated conditions.

Ln 36 What do you mean by "real soils."?

Ln 39. Please explain what you mean by this sentence, "the effect of macropores being expected to culminate when they are activated (i.e., water-saturated)" The sentence is tough to understand.

Ln 63 Please include the error in the construction of the cylindrical macropores generated by the 3D printer. Commonly for smaller diameters, the error might increase considerably.

Ln. 45. In the sentence related to the objective, please state that the soil system was under saturated conditions. The breakthrough experiment can be done either in undersaturated or saturated conditions.

Ln 110. You mentioned that MRI is utilized to visualize water flow and $Gd^{3+}$ transport. However, in this section, you only mentioned that MRI was used to visualize the latter. The transport of Gd3+ can be somewhat different from the water flow due to dispersion or diffusion. Did you visualize water flow or just $Gd^{3+}$?. In the results and discussion, you barely mentioned water flow also. Please check.

Ln 141 You mention "Stokes equation". However, I would like that you mention the specific equation that you are using. I might think that you are simulating Creeping flow or Stokes flow in your macropore for fully saturated conditions. This kind of solution is special for low Reynolds numbers. Finally, I think you drop the gravity term in Eq. 2 (please check) because you simulate in the vertical flow direction.

Ln 151 Why are you not considering dispersion in the macropore?

Ln 167 How did you compute the new mean pore diameter?

Ln 170 Please specify if that porosity was obtained from the factory or you guessed it?

Ln 172 This equation was computed under laminar flow conditions? If so, please mention it.

Ln 174 why are you using a value different from 0.57 mm for the dispersion length?
Ln 176 Do you have a wall boundary condition for the 0.5 diameter openings that connect the macropore with the matrix?.

Technical corrections

Title: I think "macropored" is not correct. Perhaps macroporous.

Ln 2. "Macropored porous media" do not sound okay to me. It could be "filled with macropores."

Ln 2-3. I suggest to change "numerically investigated, **and the**" by "numerically investigated. **The**".

Ln 4. I think you can delete "of the presence of."

Ln 5. I suggest you modify "macropored systems" by macroporous systems. Please check yourself all the times that you mentioned "macropored."

Ln. 16. I suggest deleting "Broadly speaking."

Ln 24. I suggest removing "presence of the"

Ln 30. Check that you used "and" several times. Perhaps you should remove the "and" in "and frits." Alternatively, rewrite the sentence.

Ln 33. Please considered change "may trigger disturbance of water flow" into "may trigger water flow disturbance."

Ln 36. Please considered to split this sentence into two "Real soils frequently contain macropores (Beven and Germann, 2013), which are large and continuous openings known to be involved in the rapid displacement of water and chemical substances, and various breakthrough experiments have been performed to study the role played by single macropores embedded in porous medium (Allaire et al., 2009)"

Ln 39. Please check the next sentence because is hard to understand "Unsaturated conditions being difficult to sustain in a well-controlled fashion, and the effect of macropores being expected to culminate when they are activated (i.e., water-saturated), many results have been obtained from macropored columns operated in the saturated regime, with different artificial systems: packed soils containing constructed macropores, macropored sandy media, glass bead packings crossed by a macropore, etc. (Allaire et al., 2009; Li and Ghodrati, 1997; Ghodrati et al., 1999; Lamy et al., 2009; Batany et al., 2019)." I think you should not use "being" e.g. "Unsaturated conditions **are** challenging to sustain in a well-controlled fashion. Also considered to split and rephrase the sentence is too long.

Ln 45. I suggest removing "of the presence."

Ln 58. I suggest modifying "Finally, the sand was dried at 105 ∘C during 24 h and" into "Finally, the sand was dried at 105 ∘C for 24 h and."

Ln 76 In the next sentence "for some of the experiments", please remove "of the".

Ln 114 In the next sentence, "Due to its paramagnetic properties, $Gd_{3+}$ is known to be an excellent MRI contrasting" please remove "known to be".

Ln 125. Please consider removing "as" from the next sentence "as the measured 16 echoes."

Ln 126-129 Please use the next sentence " Moreover, due to the short recycling delay used to keep measurement time below 4 min, the resulting signal depends simultaneously on the spin-lattice relaxation time T1 and the spin-spin relaxation time T2, thus complicating quantification of a simple comparison with a reference. The MRI images can nevertheless be used to evaluate where $Gd_{3+}$ is present within the column."

Ln 136. Please remove "one" from "another ONE for the macropore."

Ln 145 Please remove "in" from the next sentence "the surrounding porous medium and in the filters"

Ln 151 I suggest to modify "The transport of the non-reactive solute was modeled" into " The non-reactive solute transport was modeled".

Ln 155 Please remove IN as before.

Ln 169 Please use the next sentence (included "a") "The filter was modeled as a thin porous slab."

Ln 174 Please correct "longitudinal."

Ln 179. In the next sentence "of injected solution and to 0 afterward" it should be "afterward" and please remove "to"

Ln 205 I suggest modifying "The decrease of" into "The decrease in".

Ln 209. In the next sentence, "hollow cylinder used in column B with a plain one to investigate" replace "by" for "with".
"
Ln 211 In the next sentence, "the presence (blue curve) and in the absence." please remove "in"

Ln 214 In the next sentence, "Thus, the solute having entered the column" please replace having entered by entering.

Figure 3: In the next sentence, "the presence (blue curve) and in the absence" please remove "in"

Ln 217  Please consider changing your sentence for this one "This is not the case for column B, whose macropore is perforated; thus, it can experience some solute transfer between the macropore and the surrounding matrix"

Ln 230 Please modify "These images were taken over the course of the injection" into "These images were taken throughout the injection of"

Ln 271 Please considered modifying "which is its influence on the position of the second peak" into "which is its influence on the second peak position"

Ln 273 Please consider to change "On the downside, the modeling of the first peak" into "On the downside, the first peak modeling appears challenging"

Ln 274 Please change "we did not succeed…." by "we failed to reproduce this portion of the experimental BTCs quantitatively."

Ln 244 Plase modify the next sentence, "We start the discussion by the case" into "We start the discussion with the case"

Ln 287 I think you should change "Afterwards" by "Afterward"

Ln 293 Please change "on the way" for "how" and "transfered" by  "transferred"

Ln 301 Please modify "nearly horizontal until its gets sufficiently" into "nearly horizontal until it gets sufficiently"

Ln 372 Please consider modifying "Indeed, when it comes to fitting the experimental data, a good knowledge" by "Indeed, when it comes to fitting the experimental data, good knowledge"

Ln 331 Please modify "field were obtained at the inlet and at the outlet of the column" by "field were obtained at the inlet and the outlet of the column"

Ln 339 I think you should remove "one"

Ln 343 This sentence is too long, considered split perhaps in "whereas it" by ".In contrast, it"... "However, despite the symmetry of the streamlines between the inlet and the outlet of the column when filters are present at both extremities, qualitatively, solute transport seems to be rather unaffected by the presence of an inlet filter, whereas it is strongly impacted by the presence of the outlet filter: images acquired by MRI (Figs. 4b and 4c) and computer simulations (Figs. 5b and 5c) show that the solute front is nearly horizontal in the lower half of a macropored column containing a perforated macropore"

Ln 347 . Please modify"The presence of the inlet filter probably alter" by  "The presence of the inlet filter probably alters"

Ln 380 Please modify "This issue requires a  careful consideration of the potential impact" by "This issue requires careful consideration of the potential impact"

---

## Author Comment (AC1) · 25 Nov 2020

**We thank the reviewer for his/her comments and his/her questions. Detailed answers (in bold) are given below. All the corrections made to the manuscript appear (in blue colour) after our response.**

General Comments

The study aims to demonstrate an exit effect (outer filter) in solute transport in a soil column with artificial macropores. They utilize advanced methodologies such as breakthrough experiments, MRI, and Comsol Multiphysics software. The results showed that an exit effect occurs with a good agreement between the three independent methodologies. I agree with your findings and congratulate the researchers for the hard work. Indeed, the study shows an exit effect on solute transport. Therefore, the overall research question is answered.

**We thank the reviewer for his positive appreciation of our work.**

Before publication, I suggest an in-depth English edition. I found some technical corrections (see below); however, I am sure that I drop some more. Finally, I would like the authors to answer the next general and specific comments.

**We made the technical corrections suggested by the reviewer. We will reread very carefully the manuscript to correct residual language errors.**

Q1. You did a nice introduction to the problem. You highlight that column laboratory experiments are commonly used to study the transport of various contaminants in soils or fit experimental data with a transport model. Considering the second highlight (fit experimental data), we commonly fit the dispersion length for a non-reactive solute with BTCs without considering exit effects and then use it for model simulations. Given your results for columns B and C. Do you think that the magnitude of the fitted dispersion length using the blue or red curves in Figure 3, might be strongly different? I mean, we should be worried about the current estimation of dispersion length without considering exit effect under laboratory conditions? I think the discussion about this is important in your manuscript because you also mentioned this in the conclusions. You indeed found an exit effect with your study, but it can be irrelevant for the fitted parameter (e.g., dispersion length). Finally, it could be interesting to see the exit effect for an adsorbing solute with COMSOL to expand this study (optional).

**Indeed, we think that dispersivity (= dispersion length, if we understand correctly the terminology used by the reviewer) values inferred from BTCs should be taken with a grain of salt and can be impacted by entrance/exit effects. This statement may seem conservative, but even for homogeneous porous media, doubts have been raised about the pertinence of dispersivity values fitted from BTCs. For instance, you can have a look at Fig. 9 of [DOI: 10.1103/PhysRevE.94.053107]: in this article, the significant discrepancy between the "global" dispersivity fitted from the BTCs and the "local" one measured by NMR is explained by the existence of entrance/exit effects. As shown in our manuscript, with the macroporous systems we investigated, an exit effect is clearly visible, both on water flow and solute transport, which was not the case for homogeneous porous media. This effect induces a non-unidirectional water flow near the outlet filter and necessarily impacts the quality of estimates of hydrodynamic parameters (including dispersivity) when a one-dimensional transport model is employed. Consequently, compared to homogeneous porous media, we expect even more bias for the estimates of dispersivity with heterogeneous columns. More specifically, for**

the blue and red curves of Fig. 3, we found a twofold difference in the dispersivity values between the two scenarios (with and without the filter).

Finally, we agree that performing these kinds of experiments or computer simulations with a reactive solute would be very interesting. However, this topic is out of the scope of this work, but it could be the subject of further studies.

Q2. You demonstrate an exit effect using a soil system with a high macropore diameter (3 mm) in a sandy matrix with a mean pore diameter of 0.58 mm (for some researches, that diameter is indeed a macropore itself). The 3 mm macropore connects perfectly to the top and bottom boundary of the column with the filter. Therefore, your system allows fast water flow and solute transport in both domains, which can be different from the standard description of macropore flow where the soil matrix has very low permeability, and macropores can terminate at different depths. Thus, I think your system is convenient for observing the exit effect because of the high differences in permeability between the filter and the bulk soil system. What happens with the exit effect if the macropore terminates in the soil matrix before reach the bottom filter? What happens with smaller macropore diameters where the flow velocity is lower than for 3 mm macropores?. These questions are important for analyzing your conclusions. Please check your conclusion from line 371 onwards; they cannot be general because your study is particular, perhaps, unrealistic.

**The reviewer questions the chosen experimental conditions. Indeed, many contrasted conditions and scenarios could be tested. We promoted experimental conditions (column length, choice of the matrix, macropore diameter) allowing fast water flow through the macropore and thus reasonable experiment durations while providing convincing results.**

**The genericity of our results can be questioned at two levels: (i) relative characteristics of the macropore and the matrix and (ii) relative characteristics of the macropore and the matrix with regard to the filter.**

   i)     **Regarding the first point, the importance of the flow in the surrounding porous matrix is related to the permeability and the surface area ratios between both domains. Our system could be used with a less permeable matrix or a macropore having a larger diameter to deal with situations where the velocity contrast between the macropore and the porous matrix is more pronounced. However, studying such systems could be very time-consuming (the smaller the pore velocity in the matrix, the longer for the solute to exit the column). If the macropore terminates in the porous matrix before reaching the outlet filter, the BTC will be different [DOI: 10.1016/S0169-7722(99)00079-0]. If the length of the macropore is almost equal to the length of the porous system, the discrepancy will remain moderate. It will increase if the relative length of the macropore decreases. If a macropore with a smaller diameter is used, the two peaks will probably be closer to each other, since the pore velocities in the matrix and the macropore will be closer. They will eventually merge if the diameter of the macropore is small enough (when the diameter of the macropore tends to 0, we end up with a homogeneous system and the BTC displays a single peak in this case).**

   ii)    **Regarding the relative characteristics of the macropore and the matrix with regard to the filter, we also expect an impact. The impact of a porous layer inserted in a porous matrix is related to its hydraulic conductivity and that of the surrounding porous medium [DOI: 10.1021/es035029s]. In our case, the observed strong impact of the filter was also likely due to the gap in hydraulic conductivity between the filter and the surrounding matrix and macropore. Consequently, as**

**stated by the reviewer, we expect less influence for finer matrices and smaller macropores. However, that hypothesis requires more investigation.**

**We would like to stress that there is no canonical way of studying experimentally heterogeneous porous media. Choices have to be made. Our work is certainly not the definitive answer to this kind of studies. Our goal was simply to raise awareness on the possibly strong impact of exit effects on BTCs and to explain their origin.**

**Finally, we believe that the range of validity of our conclusions exceeds the case of the particular systems used in this work. The key point is the non-unidirectional nature of the flow field that makes it difficult to reliably infer transport properties: this is a general feature of heterogeneous porous media and it will unavoidably affect parameter inference to some extent if a 1D macroscopic transport model using lumped parameters is used. However, we agree that we did not prove this point in full generality and we rephrased our conclusion accordingly.**

Q3 Finally, in your conclusion (Ln 371), you mentioned that "Our results shows that this knowledge is crucial for the understanding of the outcome of transport experiments in heterogeneous columns and for the accurate inference of transport properties from breakthrough studies." As I mentioned in Q1, I think you did not demonstrate the last point. We do not know if the exit effect is critical for the accurate inference of transport properties from breakthrough studies with your study. From your study, we can know that the exit effect happens in your particular setup (which is probably deviated for common heterogeneous soils with macropores).

**This question is somehow related to Q1 and Q2. We agree that the quantitative importance of the exit effects on dispersivity inference is not detailed in the manuscript. However, we explicitly showed that the exit affects solute transfer and BTCs at the outlet. The analysis with any 1D macroscopic model (even when using a dual permeability approach) will fail in providing the nominal value of dispersivity of both the matrix and the macropore.**

**In addition to that, our line of thought is as follows: given an experimental BTC and a macroscopic transport model, one can infer a dispersion coefficient related to matrix transport; this coefficient is not intrinsic to the matrix since it depends on the pore velocity (see Eq. 5); consequently, the most relevant quantity is the dispersivity (= dispersion / pore velocity, if molecular diffusion can be neglected). However, our computer simulations show that the pore velocity within the matrix cannot be described by a single numerical value, as for homogeneous porous media (see Fig. 6b). Therefore, the most reasonable choice is to divide the inferred dispersion coefficient by the "bulk" pore velocity, i.e., the ratio between the length of the column and the time associated to the maximum of the second peak. However, our work shows that this time is affected by the presence/absence of an outlet filter. This is not a favourable situation for doing accurate inferences. For instance, as stated in our response to Q1, there is a twofold difference in the dispersivity values corresponding to the red (without any outlet filter) and blue curves (with an outlet filter) of Fig. 3.**

Specific comments

Abstract: Please mention that your study is only under saturated conditions.

**Done.**

Ln 36 What do you mean by "real soils."?

**We suppressed "real".**

Ln 39. Please explain what you mean by this sentence, "the effect of macropores being expected to culminate when they are activated (i.e., water-saturated)" The sentence is tough to understand.

**Macropores only impact water and solute transport when they are filled with water. This is only the case when the porous medium is water-saturated, or at least close to saturation. We rephrased this sentence to make this point clearer.**

Ln 63 Please include the error in the construction of the cylindrical macropores generated by the 3D printer. Commonly for smaller diameters, the error might increase considerably.

**The spatial resolution of the 3D printer is 0.1 mm. We added this information in the manuscript.**

Ln. 45. In the sentence related to the objective, please state that the soil system was under saturated conditions. The breakthrough experiment can be done either in undersaturated or saturated conditions.

**Done.**

Ln 110. You mentioned that MRI is utilized to visualize water flow and $Gd^{3+}$ transport. However, in this section, you only mentioned that MRI was used to visualize the latter. The transport of $Gd^{3+}$ can be somewhat different from the water flow due to dispersion or diffusion. Did you visualize water flow or just $Gd^{3+}$? In the results and discussion, you barely mentioned water flow also. Please check.

**We imaged the presence of $Gd^{3+}$with the MRI sequence used in this work. The local $Gd^{3+}$ concentration is related to water flow, but also to other processes, as noticed by the reviewer. We clarified this point in the text: in the sentence "We used these solutions for visualizing water flow and Gd3+ transport within the column by MRI", we suppressed "water flow and". Some MRI sequences have been developed to directly visualize water flow in porous media, but we did not use any of those in this work.**

Ln 141 You mention "Stokes equation". However, I would like that you mention the specific equation that you are using. I might think that you are simulating Creeping flow or Stokes flow in your macropore for fully saturated conditions. This kind of solution is special for low Reynolds numbers. Finally, I think you drop the gravity term in Eq. 2 (please check) because you simulate in the vertical flow direction.

**The Stokes equations are written in Eq. 2. They are indeed used do model a Stokes flow (also called creeping flow) and are valid for moderate Reynolds numbers. The flow of a Newtonian liquid within a capillary (like a macropore) remains laminar for Re $\leq$ 2.1×10³ [DOI: 10.1126/science.1203223], which is indeed the case in our experiments.**

**For the gravity term, we thank the reviewer for pointing out this omission in the text. We made the appropriate correction in the manuscript.**

Ln 151 Why are you not considering dispersion in the macropore?

**Dispersion is an emerging (i.e. macroscopic) concept stemming from the combined effect of the water velocity field and molecular diffusion. At small scale, the sole physical processes are advection and diffusion.**

**When one deals with a soil column or even a homogeneous porous medium, the pore-scale velocity field is very complicated and it is not possible nor desirable to disentangle the advective and diffusive components of transport. In this case, the effect of small-scale variability on macroscopic solute transport is described statistically, with a longitudinal dispersion coefficient or a dispersion tensor. But in some situations, for instance when the flow domain is simple enough (e.g. for a capillary), advection and diffusion can be taken explicitly into account to model solute transport (the standard example is tracer transport in laminar flow through a straight tube of circular cross-section, which leads to Taylor-Aris dispersion for long enough tubes).**

**This is what we have done in the macropore: we solved the Stokes and the convection-diffusion equation in this domain and it is the interplay between advection and diffusion that generates some Taylor dispersion.**

Ln 167 How did you compute the new mean pore diameter?

**This is the mean diameter of the grains $d_g$ (related but not strictly equal to the mean pore diameter). It was estimated by modifying the permeability of the porous matrix to get a good fit of the experimental BTCs and by using the Kozeny-Carman equation to get $d_g$.**

Ln 170 Please specify if that porosity was obtained from the factory or you guessed it?

**We estimated it. Its precise value does not have much effect on the numerical results.**

Ln 172 This equation was computed under laminar flow conditions? If so, please mention it.

**Yes. We mentioned it in the revised version of the manuscript.**

Ln 174 why are you using a value different from 0.57 mm for the dispersion length?

**Actually, we used 0.38 mm, which is the median pore diameter measured by X-ray tomography (see Sec. 2.1).**

Ln 176 Do you have a wall boundary condition for the 0.5 diameter openings that connect the macropore with the matrix?.

**No, as it would forbid water and solute exchange between the macropore and the matrix. The lateral boundary of the perforated macropore inserted in column B was not modelled. We considered that the interior of the macropore and the matrix were directly in contact. From a numerical point of view, the Brinkman and Navier-Stokes domains were thus contiguous.**

Technical corrections

Title: I think "macropored" is not correct. Perhaps macroporous.

**We followed the suggestion of the reviewer.**

Ln 2. "Macropored porous media" do not sound okay to me. It could be "filled with macropores."

**We rephrased that sentence.**

Ln 2-3. I suggest to change "numerically investigated, **and the**" by "numerically investigated. **The**".

**Done.**

Ln 4. I think you can delete "of the presence of."

**Done.**

Ln 5. I suggest you modify "macropored systems" by macroporous systems. Please check yourself all the times that you mentioned "macropored."

**Done. We made the corresponding modifications everywhere in the manuscript.**

Ln. 16. I suggest deleting "Broadly speaking."

**Done.**

Ln 24. I suggest removing "presence of the"

**Done.**

Ln 30. Check that you used "and" several times. Perhaps you should remove the "and" in "and frits." Alternatively, rewrite the sentence.

**We suppressed the first "and".**

Ln 33. Please considered change "may trigger disturbance of water flow" into "may trigger water flow disturbance."

**Done.**

Ln 36. Please considered to split this sentence into two "Real soils frequently contain macropores (Beven and Germann, 2013), which are large and continuous openings known to be involved in the rapid displacement of water and chemical substances, and various breakthrough experiments have been performed to study the role played by single macropores embedded in porous medium (Allaire et al., 2009)"

**This sentence has been split in two.**

Ln 39. Please check the next sentence because is hard to understand "Unsaturated conditions being difficult to sustain in a well-controlled fashion, and the effect of macropores being expected to

culminate when they are activated (i.e., water-saturated), many results have been obtained from macropored columns operated in the saturated regime, with different artificial systems: packed soils containing constructed macropores, macropored sandy media, glass bead packings crossed by a macropore, etc. (Allaire et al., 2009; Li and Ghodrati, 1997; Ghodrati et al., 1999; Lamy et al., 2009; Batany et al., 2019)." I think you should not use "being" e.g. "Unsaturated conditions **are** challenging to sustain in a well-controlled fashion. Also considered to split and rephrase the sentence is too long.

**We rephrased that part.**

Ln 45. I suggest removing "of the presence."

**Done.**

Ln 58. I suggest modifying "Finally, the sand was dried at 105 ∘C during 24 h and" into "Finally, the sand was dried at 105 ∘C for 24 h and."

**Done.**

Ln 76 In the next sentence "for some of the experiments", please remove "of the".

**Done.**

Ln 114 In the next sentence, "Due to its paramagnetic properties, $Gd^{3+}$ is known to be an excellent MRI contrasting" please remove "known to be".

**Done.**

Ln 125. Please consider removing "as" from the next sentence "as the measured 16 echoes."

**We rephrased that part.**

Ln 126-129 Please use the next sentence " Moreover, due to the short recycling delay used to keep measurement time below 4 min, the resulting signal depends simultaneously on the spin-lattice relaxation time T1 and the spin-spin relaxation time T2, thus complicating quantification of a simple comparison with a reference. The MRI images can nevertheless be used to evaluate where $Gd_{3+}$ is present within the column."

**Done.**

Ln 136. Please remove "one" from "another ONE for the macropore."

**Done.**

Ln 145 Please remove "in" from the next sentence "the surrounding porous medium and in the filters"

**Done.**

Ln 151 I suggest to modify "The transport of the non-reactive solute was modeled" into " The non-reactive solute transport was modeled".

**Done.**

Ln 155 Please remove IN as before.

**Done.**

Ln 169 Please use the next sentence (included "a") "The filter was modeled as a thin porous slab."

**Done.**

Ln 174 Please correct "longitudinal."

**Done.**

Ln 179. In the next sentence"of injected solution and to 0 afterward" it should be "afterward" and please remove "to"

**Done.**

Ln 205 I suggest modifying "The decrease of" into "The decrease in".

**We believe both formulations are correct. We kept the original one.**

Ln 209. In the next sentence, "hollow cylinder used in column B with a plain one to investigate" replace "by" for "with".

**Done, even if we think that "by" and "with" can be used interchangeably in this case.**

Ln 211 In the next sentence, "the presence (blue curve) and in the absence." please remove "in"

**Done.**

Ln 214 In the next sentence, "Thus, the solute having entered the column" please replace having entered by entering.

**Done.**

Figure 3: In the next sentence, "the presence (blue curve) and in the absence" please remove "in"

**Done.**

Ln 217 Please consider changing your sentence for this one "This is not the case for column B, whose macropore is perforated; thus, it can experience some solute transfer between the macropore and the surrounding matrix"

**Done.**

Ln 230 Please modify "These images were taken over the course of the injection" into "These images were taken throughout the injection of"

**Done.**

Ln 271 Please considered modifying "which is its influence on the position of the second peak" into "which is its influence on the second peak position"

**Done.**

Ln 273 Please consider to change "On the downside, the modeling of the first peak" into "On the downside, the first peak modeling appears challenging"

**We prefer the initial formulation.**

Ln 274 Please change "we did not succeed…." by "we failed to reproduce this portion of the experimental BTCs quantitatively."

**Done.**

Ln 244 Plase modify the next sentence, "We start the discussion by the case" into "We start the discussion with the case"

**Done.**

Ln 287 I think you should change "Afterwards" by "Afterward"

**Done.**

Ln 293 Please change "on the way" for "how" and "transfered" by "transferred"

**Done.**

Ln 301 Please modify "nearly horizontal until its gets sufficiently" into "nearly horizontal until it gets sufficiently"

**Done.**

Ln 372 Please consider modifying "Indeed, when it comes to fitting the experimental data, a good knowledge" by "Indeed, when it comes to fitting the experimental data, good knowledge"

**Done.**

Ln 331 Please modify "field were obtained at the inlet and at the outlet of the column" by "field were obtained at the inlet and the outlet of the column"

**Done.**

Ln 339 I think you should remove "one"

**We inserted a "that" between the two "one".**

Ln 343 This sentence is too long, considered split perhaps in "whereas it" by ".In contrast, it"... "
[revised manuscript text omitted]

---

## Referee Comment (RC2) · Anonymous Referee #2 · 26 Nov 2020

Manuscript HESS-2020-494 investigates the impact of a filter located at the end of a sand column with and without macropores on the flow field under steady-state transport experiments with inert solutes. The authors demonstrate that the filter modifies the flow field substantially in case that there are specific preferential flow paths present in the porous media. Bringing attention to this fact is important to avoid and address similar pitfalls in future experiments on water flow and solute transport through soils and rocks. It is therefore of interest to the readership of HESS.

I am thanking the authors for having delivered a very well developed manuscript. It is well structured and the English is on a very good level. The editor should note that I am not an expert in MRI and cannot say much about the description of the MRI setup other than that it looks like a typical description of an MRI setup (which still is a bit like

magic for me). The equations in the manuscript appear to be correct to me, but I have to admit that I have no idea how equation 7 is derived and do not have access to Bruus (2007). In the following recommendation, I am assuming that everything is sound.

I only have a few general and specific remarks that the authors need to address carefully before the manuscript can be published. These are however minor.

General comment:

The impact of a filter under upward flow conditions is very similar to the presence of a seepage face at the bottom of soil columns or lysimeters that are open to the atmosphere (see Flury et al., 1999 in the reference list below). Your study implies that also the installation of a porous plate at the lysimeter's lower boundary will not result in a flow-field that is similar to the one that would be observed in the field. The authors might want to make this point, too. If experiments are conducted under unsaturated conditions on lysimeters or soil columns, the distortion of the flow field is expected to be even more extreme than under saturated conditions, because also the water content changes in the vicinity to the outlet. However, the phenomenon observed in HESS-2020-494 is also observed at a seepage face under (near-) saturated conditions as the study of Koestel and Larsbo (2014) shows.

Specific comments:

L2 and following uses of the term: "macropored porous media" is not used in the soil hydrology and solute transport community. I recommend using "porous media with (artificial) macropores" instead.

L43-44: this is not entirely true. See Flury et al. (1999) for discussions on the effect of the column exit and Kreft and Zuber (1978) for the effect of the entrance.

L53: please use past tense when describing your experiment

L61-62: Pore diameters derived from X-ray images are strongly influenced by the image resolution and image processing steps chosen to arrive at a binary image of the

pore space. Furthermore, the average pore size from X-ray images will always be over-estimated, because the smaller pores are not visible in the image. If you mention the X-ray measurements, you need to elaborate on how you arrived at the pore diameter measurement. However, since you fitted the effective pore diameter anyway later on, so that it results in a good fit of the breakthrough curves, I recommend to remove all references to the X-ray measurements from the manuscript.

L94: conservative tracers

L97: better: "The columns were installed in an upright position.."

L126 the measurement time

Equations 2, 3, and 4: please decide for one way of writing the Laplace operator

L164: . . .Kozeny-Carman equation, assuming perfectly spherical sand grains.

L169: "holes of length" Should it not be: "holes of diameter"?

L174: please provide a justification for your choice of dispersivity and tortuosity

L223-227: Nice!

L285-286: the solute remains more concentrated within the macropore

L304: not the "presence of the filter" implies this but "the transport patterns in the presence of a filter"

L343-345: I do not agree. I am sure you would see a substantial effect of the inlet filter if you would model the transport without such a filter (see Kreft and Zuber, 1978).

L351-365: please remove. It repeats what has been very explicitly discussed in the section before.

Figure 4: please use a color scheme or invert the gray-scale. At the moment, the tracer plumes are difficult to discern.

References

Flury, M., Yates, M.V., Jury, W.A., 1999. Numerical analysis of the effect of the lower boundary condition on solute transport in lysimeters. Soil Sci. Soc. Am. J. 63(6), 1493-1499.

Koestel, J., Larsbo, M., 2014. Imaging and quantification of preferential solute transport in soil macropores. Water Resour. Res. 50(5), 4357-4378.

Kreft, A., Zuber, A., 1978. On the physical meaning of the dispersion equation and its solutions for different initial and boundary conditions. Chemical Engineering Science 33(11), 1471-1480.

---

## Author Comment (AC2) · 30 Nov 2020

**We thank the referee for his/her comments and his/her questions. Detailed answers (in bold) are given below. All the corrections made to the manuscript appear (in red colour) after our response.**

Manuscript HESS-2020-494 investigates the impact of a filter located at the end of a sand column with and without macropores on the flow field under steady-state transport experiments with inert solutes. The authors demonstrate that the filter modifies the flow field substantially in case that there are specific preferential flow paths present in the porous media. Bringing attention to this fact is important to avoid and address similar pitfalls in future experiments on water flow and solute transport through soils and rocks. It is therefore of interest to the readership of HESS.

**We are glad to learn that you appreciated our manuscript. We also think it may be of interest to a rather broad readership.**

I am thanking the authors for having delivered a very well developed manuscript. It is well structured and the English is on a very good level. The editor should note that I am not an expert in MRI and cannot say much about the description of the MRI setup other than that it looks like a typical description of an MRI setup (which still is a bit like magic for me). The equations in the manuscript appear to be correct to me, but I have to admit that I have no idea how equation 7 is derived and do not have access to Bruus (2007). In the following recommendation, I am assuming that everything is sound.

**Regarding Eq. 7, it stems from the analytical solution of the Navier-Stokes equation for the velocity field in a channel having a rectangular cross-section. Contrary to the circular cross-section case, there is no simple closed-form formula for the velocity field in this situation. However, it can be expressed as the sum of a Fourier series and Eq. 7 can be readily deduced from this Fourier expansion.**

I only have a few general and specific remarks that the authors need to address carefully before the manuscript can be published. These are however minor.

General comment:

The impact of a filter under upward flow conditions is very similar to the presence of a seepage face at the bottom of soil columns or lysimeters that are open to the atmosphere (see Flury et al., 1999 in the reference list below). Your study implies that also the installation of a porous plate at the lysimeter's lower boundary will not result in a flow-field that is similar to the one that would be observed in the field. The authors might want to make this point, too. If experiments are conducted under unsaturated conditions on lysimeters or soil columns, the distortion of the flow field is expected to be even more extreme than under saturated conditions, because also the water content changes in the vicinity to the outlet. However, the phenomenon observed in HESS-2020-494 is also observed at a seepage face under (near-) saturated conditions as the study of Koestel and Larsbo (2014) shows.

**We thank the referee for this suggestion. We agree about the potentially substantial effect of a seepage face / layer on water flow under unsaturated conditions. Indeed, under unsaturated conditions, the seepage face and the layer acts as a barrier to the flow and**

**triggers an increase in water content close to the lower boundary. Such an increase is expected to promote dilution, and to impact flow and solute pathways. Thus, it can affect solute breakthrough at the column outlet. We added the references indicated by the referee in the revised version of the manuscript.**

Specific comments:

L2 and following uses of the term: "macropored porous media" is not used in the soil hydrology and solute transport community. I recommend using "porous media with (artificial) macropores" instead.

**We have followed the suggestion of Anonymous Referee #1 and used "macroporous" instead of "macropored".**

L43-44: this is not entirely true. See Flury et al. (1999) for discussions on the effect of the column exit and Kreft and Zuber (1978) for the effect of the entrance.

**We modified this sentence. We replaced**
*"However, to our knowledge, the potential impact of exit effects on solute transfer through macropored porous media has never been investigated so far."*
**by**
*"The impact of entrance/exit effects has been noticed in a few studies (Flury et al., 1999; Kreft and Zuber, 1978), but to our knowledge, the underlying mechanisms responsible for these effects have never been thoroughly investigated so far."*

L53: please use past tense when describing your experiment

**Done.**

L61-62: Pore diameters derived from X-ray images are strongly influenced by the image resolution and image processing steps chosen to arrive at a binary image of the pore space. Furthermore, the average pore size from X-ray images will always be overestimated, because the smaller pores are not visible in the image. If you mention the X-ray measurements, you need to elaborate on how you arrived at the pore diameter measurement. However, since you fitted the effective pore diameter anyway later on, so that it results in a good fit of the breakthrough curves, I recommend to remove all references to the X-ray measurements from the manuscript.

**As stated by the referee, the median pore diameter inferred from X-ray tomography measurements does not lie at the heart of our study. We thus followed his/her recommendation and removed from the manuscript all the elements related to these measurements.**

L94: conservative tracers

**Done.**

L97: better: "The columns were installed in an upright position..."

**Done.**

L126 the measurement time

**Done.**

Equations 2, 3, and 4: please decide for one way of writing the Laplace operator

**There was a typo in the diffusion term of Eq. 4. When dispersion (or even diffusion) is not a scalar but a second-order tensor, there is no Laplacian in the diffusion term but nested differential operators (divergence of (dispersion tensor × gradient of concentration)). It is only when the dispersion tensor is isotropic that this term is reduced to (dispersion coefficient × Laplacian of concentration). We corrected that typo.**

L164: …Kozeny-Carman equation, assuming perfectly spherical sand grains.

**Done.**

L169: "holes of length" Should it not be: "holes of diameter"?

**These holes have a square shape, not a circular one. We wrote "holes of side length" instead of "holes of length".**

L174: please provide a justification for your choice of dispersivity and tortuosity

**We inferred the dispersivity of the Hostun sand from the BTCs measured with column A (homogeneous control column) and we used this value for the porous matrix of column B. This information has been included in the revised version of the manuscript. As for the dispersivity of the filter, we chose to set it equal to the characteristic length of the pores of the filter (10 µm); since the filter is very thin, its precise value has no sensible impact on the numerical BTCs.**

**Regarding the tortuosity, for homogeneous, unconsolidated and water-saturated sphere packings, it is known to be approximately equal to 0.7 (Liasneuski et al., 2014). However, retaining a precise value would have been useless in this study because, for the flow rate we investigated, the value of the tortuosity does not affect the numerical results we obtained. This is because in the conditions of our study, the diffusion component $\tau\, D_0$ of the dispersion coefficient is much smaller than the mechanical dispersion term ($\lambda\, |u|$) in Eq. 5.**

L223-227: Nice!

**Thank you!**

L285-286: the solute remains more concentrated within the macropore

**Done.**

L304: not the "presence of the filter" implies this but "the transport patterns in the presence of a filter"

**Done.**

L343-345: I do not agree. I am sure you would see a substantial effect of the inlet filter if you would model the transport without such a filter (see Kreft and Zuber, 1978).

**We think that we agree with the referee. We wrote that "solute transport seems to be rather unaffected by the presence of an inlet filter (…)": what we meant in that, both in the MRI images and the numerical concentration maps, the solute front does not seem much affected by the inlet filter (or at least, to a far lesser extent than in the vicinity of the outlet filter). However, it does not mean that everything is perfectly fine near the inlet filter and that this boundary device has no impact on solute dispersion. Actually, we even think that the contrary is true: Kreft and Zuber (1978), as well as other articles such as Lehoux et al. (2016), Bromly et al. (2007), Parker and Albrecht (1987), or James and Rubin (1972), support this assertion. However our study is not conclusive on this point. Subtle inlet effects (subtle in the sense that they are not straightforwardly discernible in the MRI images or the concentration maps) probably exist, but the MRI and numerical data we have obtained neither rule out nor support their existence. We tried to make this point clearer in the revised version of the manuscript.**

L351-365: please remove. It repeats what has been very explicitly discussed in the section before.

**We suppressed a substantial fraction of the section mentioned by the referee.**

Figure 4: please use a color scheme or invert the gray-scale. At the moment, the tracer plumes are difficult to discern.

**We tried various colour schemes in an attempt to improve the discernibility of the plume, without achieving a better result, at least in our opinion. Four tries are reproduced below for the image PV$_2$ of Fig. 4b. Some colour schemes probably look better than a simple grayscale, but we do not find that they improve the readability of the images. This is obviously a question of visual perception but we are unsure on how to deal with it adequately. We added some information about the significance of the colour scheme in the caption of Fig. 4.**

[revised manuscript text omitted]